# Genome sequence-based identification of *Enterobacter* strains and description of *Enterobacter pasteurii* sp. nov.

Praveen Rahi,[1] Estelle Mühle,[1] Cyril Scandola,[2] Gerald Touak,[1] Dominique Clermont[1]

**ABSTRACT**    Members of the *Enterobacter cloacae* complex (ECC) are almost ubiquitous in nature and can act as pathogens. In this study, we used a polyphasic taxonomy approach to establish the accurate taxonomic position of six strains within the ECC. Notably, the 16S rRNA gene-based phylogeny failed to group all *Enterobacter* species into a monophyletic cluster. As an alternative to this, we explored genome sequence-based phylogenetic approaches. The bac120 gene-based phylogeny successfully grouped all *Enterobacter* species into a monophyletic cluster, although some species-level clustering conflicted with average nucleotide identity (ANI) values. Furthermore, the *Enterobacter*-specific core gene phylogeny resolved all species, aligning with ANI results. Three strains were identified as *Enterobacter asburiae*, while strain P99 was classified as "*Enterobacter xiangfangensis*" and strain C45 as *Enterobacter quasihormaechei*. Conversely, strain A-8[T] formed a distinct cluster in all phylogenies, with ANI and digital DNA-DNA hybridization (dDDH) values below the species threshold (<92% and <44%, respectively) with all known *Enterobacter* species. Matrix-assisted laser desorption/ionization time-of-flight mass spectrometry biotyper results confirmed that all strains belonged to the ECC. However, the comparison of mean spectrum profile of strain A-8[T] with those of other strains revealed the presence of 18 unique peaks, highlighting its distinct protein profile. Based on comprehensive genotypic and phenotypic characterizations, we propose that strain A-8[T] is a new species of the genus *Enterobacter*, which is named *Enterobacter pasteurii* sp. nov. The type strain is A-8[T] (CIP 103550[T]; ATCC 23355[T]; DSM 26481[T]; and WDCM 00082[T]). This study advances our understanding of the ECC, emphasizing the need for multidimensional taxonomic techniques and contributing to the better management of microbial resource centers.

**IMPORTANCE**    Accurate taxonomy is essential for microbial biological resource centers, since the microbial resources are often used to support new discoveries and subsequent research. Here, we used genome sequence data, alongside matrix-assisted laser desorption/ionization time-of-flight mass spectrometer biotyper-based protein profiling, to accurately identify six *Enterobacter cloacae* complex strains. This approach effectively identified distinct species within the *E. cloacae* complex, including *Enterobacter asburiae*, "*Enterobacter xiangfangensis*," and *Enterobacter quasihormaechei*. Moreover, the study revealed the existence of a novel species within the *Enterobacter* genus, for which we proposed the name *Enterobacter pasteurii* sp. nov. In summary, this study demonstrates the significance of adopting a genome sequence-driven taxonomy approach for the precise identification of bacterial strains in a biological resource center and expands our understanding of the *E. cloacae* complex.

**KEYWORDS**    *Enterobacter*, *Enterobacteriaceae*, MALDI-TOF MS, phylogenomic, taxonomy

Address correspondence to Praveen Rahi, prahi@pasteur.fr.

The authors declare no conflict of interest.

See the funding table on p. 14.

Microbial biological resource centers (mBRCs) ensure access to authenticated and quality-controlled microbial resources, to support new discoveries and follow-on studies. With the rapid advancement in tools and techniques used in microbial taxonomy, changes in the classification of microbial strains have become inevitable. Next-generation sequencing technologies have made genome sequencing accessible and affordable and provide more accurate and reliable information for taxonomy.

The Collection of Institut Pasteur (CIP) is one of the oldest culture collections and started preserving bacterial strains as early as 1891. At present it holds a collection of more than 25,000 bacterial strains, encompassing 5,500 different species. Many of the strains within the CIP collection were deposited prior to the advent of molecular methods. Consequently, there is a need for a contemporary taxonomic validation process employing state-of-the-art methodologies for such strains. Moreover, the taxonomical complexity of recent deposits has resulted in some strains being grouped within species complexes without clear species assignment.

The focus of this study is the *Enterobacter cloacae* complex (ECC), which comprises several difficult-to-discriminate species of *Enterobacter* (1). ECC members are known to include nosocomial pathogens responsible for infections such as pneumonia, septicemia, and urinary tract infections (2). Additionally, *Enterobacter* is part of the ESKAPE group and also contains several multidrug-resistant pathogens (3).

Over the years, the taxonomy of the genus *Enterobacter* has undergone several revisions since its initial description in 1960 (4). Currently, the list of prokaryotic names with standing in nomenclature recognizes 23 species with valid and correct names (5). Core gene-based phylogenetic analyses have led to substantial taxonomic corrections, involving the reclassification of all *Enterobacter* subspecies assignments and the proposal of new species, such as *Enterobacter wuhouensis*, *Enterobacter quasihormaechei*, *Enterobacter quasiroggenkampii*, and *Enterobacter quasimori* (6, 7). Additionally, 14 novel tentative *Enterobacter* genomospecies have been identified (7), though formal proposals on their description are yet to be made.

The primary objective of this study was to employ genome sequence-based identification for six ECC strains from the CIP, deposited by various researchers between 1967 and 2018. Among these strains, three nitrate reduction-deficient mutant strains were initially deposited as "*Aerobacter aerogenes*" (8). The remaining strains consisted of beta-lactamase-producing *E. cloacae* (*Aerobacter cloacae*) strain P99 (CIP 79.28) (9) and strain A-8[T] (CIP103550[T]) (10) and a carbapenem-resistant strain C45 (CIP 111614) (11).

Previous strain identification has relied on 16S rRNA gene sequencing, which did not provide sufficient discrimination for closely related species within the *Enterobacteriaceae* family, particularly within the ECC (1, 12). Therefore, recently, genome sequencing has become a tool of choice to achieve accurate taxonomic assignments for members of the *Enterobacteriaceae* family, and researchers also used it for the members of ECC (6, 7, 11–13). However, the underutilization of genome sequence data has often led to inaccurate taxonomy assignments particularly at species level. In this study, we analyzed different features of genome sequences within the *E. cloacae* complex and established species-level taxonomy assignment to avoid any ambiguities or misperceptions in strain identification. Additionally, we used the genome sequencing data for the identification of antimicrobial resistance genes, virulence factors, and species-specific unique gene clusters. By doing so, we contribute to the establishment of a genome sequence-based approach for accurate taxonomy assignments for the members of *E. cloacae* complex and dependable curation of microbial resources.

## MATERIALS AND METHODS

### Isolation and identification

The strains of ECC were obtained from the CIP. Details on their isolation source and properties are presented in Table 1. All the cultures were grown on trypticase soy agar (TSA) with pH 7.0 and at 37°C, unless mentioned otherwise.

### 16S rRNA gene sequencing and phylogeny

For 16S rRNA sequencing, DNA was extracted from the strains using the InstaGenMatrix (Bio-Rad, USA) and stored at −20°C. The 16S rRNA gene was amplified using primers 27F 5′-AGAGTTTGATYMTGGCTCAG-3′ and 1391R 5′-GACGGGCGGTGWGTRCA-3′ and GoTaq DNA Polymerase (Promega, USA). The PCR was performed as follows: initial denaturation at 95°C for 3 min and then 30 cycles of denaturation (95°C, 45 s), annealing (60°C, 45 s), and elongation (72°C, 90 s). The amplified product was sent for Sanger sequencing at Eurofins Genomics (Germany). The sequence quality was checked with BioEdit 7.2.5, and trimming and assembly were performed using CLC Genomics Workbench 20.0.4.

Pairwise sequence similarities to closest related strains and type strains were obtained using the NCBI nucleotide database using BLASTn search tool and EZBioCloud database (14), respectively. Sequences of 16S rRNA gene from closely related species resulted from EZBioCloud search and the additional strains resulting from NCBI database search were used to construct phylogenetic trees. All sequences were aligned using MUSCLE (15), and maximum likelihood trees were built using IQ-TREE 2.2.2.2 (16) with ModelFinder (17) and 1,000 ultrafast bootstrap replications.

### Genome sequencing and phylogenetic analyses

For genome sequences, DNA was extracted using the Wizard Genomic DNA purification kit (Promega, USA) and stored at −20°C. The sequencing was performed using a NextSeq 500 instrument (Illumina, USA) with a 2 × 150 nt paired-end protocol at the Mutualized Platform for Microbiology (P2M) of the Institut Pasteur. The reads were assembled using the pipeline fq2dna 21.06 (https://gitlab.pasteur.fr/GIPhy/fq2dna). Briefly, *de novo* assemblies were done with SPAdes 3.15.2 (18), and then standard HTS read-processing steps were carried out with different tools like AlienTrimmer 2.0 (19) for trimming and clipping or Musket 1.1 (20) for error corrections. Genomic characteristics and quality were evaluated with contig_info 2.1 and checkM 1.1.3 (21). The genome sequences were annotated using prokka (22). Identification of virulence factors and antibiotic-resistant genes was done by using ABRicate (https://github.com/tseemann/abricate).

A data set of genomes was prepared including the genome assemblies of all type strains of *Enterobacter* downloaded from NCBI genome assembly database (accessed on 09 April 2023). To find genome of closely related strains, *recA* gene sequence of strain A-8$^T$ was used, a search query against RefSeq genome database of *Enterobacteriaceae* group (taxid: 543). In addition to this, genomes of type strains of a few key species from *Enterobacteriaceae* group were also included in this data set.

The core gene phylogeny was constructed using GTDBTk 2.1.1 pipeline to identify, extract, and align bac120 marker genes (23). Another phylogenetic analysis was carried out by using the core genes identified and extracted from only *Enterobacter* genome data set using Panaroo 1.3.0 (24). The maximum likelihood phylogenetic trees were inferred from the bac120 genes and *Enterobacter* core genes using IQ-TREE 2.2.2.2 (16) with ModelFinder (17) and 1,000 ultrafast bootstrap replications. The average nucleotide identity (ANI) was determined using FastANI (25) with the default settings (kmer = 16; fragment length = 3,000; and minimum shared fraction = 0.2). The core gene phylogenetic trees were displayed using iTOL (26), and ANI values were added as colored stripes. Digital DNA-DNA hybridization (DDH) values and confidence intervals were calculated using the recommended settings of the Genome-to-Genome Distance Calculator (GGDC) 2.1 (27).

TABLE 1   Details on the source of isolation and other properties of *E. cloacae* complex strains included in this study

| Strain (CIP accession) | Source of isolation | History | Other collection no. | Additional information (reference) |
|---|---|---|---|---|
| A-8[T] (CIP103550[T]) | Unknown | <- R. Mercier, bioMérieux, La Balme-les-Grottes; <- ATCC; ATCC 23355 <- Dept. Medical Microbiol, Stanford Univ. | ATCC 23355; CCUG 33777; CECT 5075; WDCM 00082; DSM 26481; NCTC 13380 | Quality control strain according to ISO/CD 11133:2009. Produces beta-lactamase (10) and also used to study microbe-microbe interactions (13) |
| P99 (CIP79.28) | Human clinical sample | <- 1979, Y.A. Chabbert, Inst. Pasteur, Paris, France, *Enterobacter cloacae*: strain P99 | NCIMB 12091 | Produces beta-lactamase (9) |
| M1_L III-I (CIP66.36) | Unknown | <- 1967, F. Pichinoty, CNRS, Marseille, France, as *Enterobacter aerogenes* "*Enterobacter cloacae*" | - | Mutant one from *Enterobacter aerogenes* L III-I, no nitrate reductase A, Gaz-negative (8) |
| M4_L III-I (CIP66.37) | Unknown | <- 1967, F. Pichinoty, CNRS, Marseille, France, as *Enterobacter aerogenes* "*Enterobacter cloacae*" | - | Mutant four from *Enterobacter aerogenes* L III-I, no nitrate reductase A, Gaz-negative (8) |
| RM4_L III-I (CIP66.39) | Unknown | <- 1967, F. Pichinoty, CNRS, Marseille, France, as *Enterobacter aerogenes* "*Enterobacter cloacae*" | - | Reverse mutant 4R from *Enterobacter aerogenes* L III-I (8) |
| C45 (CIP111614) | Human urine | <- 2018, R. Beyrouthy, Clermont Ferrand Hosp., Clermont Ferrand, France: strain C45 | - | Resistant to carbapenems (VIM-4) and possibly to colistin (mcr-9) due to IncHI2 plasmid (11) |

For comparative genome analyses, the genes unique to the members of new species were identified from the output of the pangenome analysis performed by using Panaroo 1.3.0 (24). Additionally, the core, group-specific and singleton gene clusters were identified and displayed by using anvi-display-pan program of the anvi'o (analysis and visualization) platform (28).

## MALDI-TOF MS-based identification and MSP creation

Whole-cell proteins were extracted using ethanol/formic acid after 24-h growth on TSA, to generate the mean spectral profile (MSP). The protein profiles were generated for each strain ranging from 2 to 20 kDa by matrix-assisted laser desorption/ionization time-of-flight mass spectrometer (MALDI-TOF MS) Biotyper sirius (Bruker Daltonik GmbH, Germany) (29). A total of 27 replicate spectra were generated for each strain and used to create MSP (29, 30). A MSP dendrogram was constructed using the peak list data of newly generated MSPs and from the MSPs of several other *Enterobacteriaceae* members present in the biotyper database (v12.0).

## Phenotypic features of strain A-8[T]

Morphological features of strain A-8[T] were observed by growing it on TSA incubated under aerobic conditions at 37°C for 24 h. Gram staining was performed by using Aerospray Gram (ELITechGroup, France). For negative staining, the bacterial cells were adsorbed for 10 min on 200-mesh Formvar/carbon-coated copper (Electron Microscopy Sciences, UK) glow discharged grids. The grids with cells were then washed several times on ultrapure water droplets and fixed for 10 min on a droplet of 1% glutaraldehyde in 1× PHEM buffer (60-mM PIPES, 25-mM HEPES, 10-mM EGTA, 2-mM MgCl2, and pH 7.3). After washing on ultrapure water droplets, grids were stained for 10 s with 2% aqueous uranyl acetate and dried. Transmission electron microscopy (TEM) images were captured with Tecnai Spirit 120-Kv TEM equipped with a bottom-mounted Eagle 4k × 4k camera (FEI, USA).

For scanning electron microscopy (SEM), bacterial cells were fixed with 2.5% glutaraldehyde in culture media for 1 h at room temperature and deposited on 0.01%

poly-L-lysine-coated 12-mm round glass slides for overnight sedimentation. All samples were then washed in 1× PHEM buffer (60-mM PIPES, 25-mM HEPES, 10-mM EGTA, 2-mM MgCl2, and pH 7.3), postfixed in 1% osmium tetroxide for 1 h, washed in water, and dehydrated in ethanol, increasing series from 25% to 100%. Samples were desiccated using a critical point dryer (Leica), mounted on SEM stubs and sputtered with 20-nm gold/palladium. Bacterial cells were observed using a SEM IT700HR (JEOL) at 5 kV with a secondary electron detector. A loopful culture was inoculated to semi-solid mannitol agar to check the motility (31).

The optimum growth temperature was determined by growing the strains on TSA plates incubated at different temperatures: 4℃, 15℃, 28℃, 37℃, and 45℃. Tryptone soy broth with different pH (4, 5.5, 7, 8.5, 10, 11, and 12) set by adding HCl or NaOH before autoclaving was used to evaluate the growth in different pH. The growth of the cultures was determined by measuring absorbance at 600 nm with the Infinite M Nano+ (TECAN, Switzerland). Growth in microaerophilic and anaerobic conditions was tested using round jars of 2.5 L with CampyGen and AnaeroGen products (ThermoFisher, USA).

Oxidase strip (Sigma-Aldrich, USA) was used to test oxidase activity, and catalase activity was done by recording the bubble formation in a 3% (*v/v*) hydrogen peroxide solution (Sigma-Aldrich, USA). Several other biochemical characteristics including enzyme activities and oxidation and reduction of carbon sources were preformed using the API 50, API ZYM and BIOTYPE 100 (bioMérieux, France) following manufacturer's instructions. The antibiotic susceptibility was determined using the disc diffusion method on Mueller-Hinton agar following the recommendations of Comité de l'antibiogramme de la Société Française de Microbiologie (CA-SFM)/European Committee on Antimicrobial Susceptibility Testing (EUCAST) (32).

## RESULTS AND DISCUSSION

### 16S rRNA gene phylogeny

The 16S rRNA gene sequences of all strains were obtained by Sanger method with a length of more than 1,400 bp. All strains showed more than 99% sequence similarity to two or more species of ECC. Based on 16S rRNA gene sequence similarity values, it was impossible to assign the strains to any particular species. Furthermore, there is growing evidence that 16S rRNA gene sequence provides low resolution to discriminate closely related species of genera such as *Enterobacter*, *Klebsiella*, *Citrobacter*, *Rhizobium*, and several others (1, 12, 33). Although the phylogenetic tree based on 16S rRNA gene sequences placed all the strains along with *Enterobacter* species (Fig. S1), we found that some species of *Enterobacter* were clustered in different groups and some of these groups often include members of other genera. Therefore, based on the 16S rRNA gene phylogeny, the taxonomic assignments of these strains remained inconclusive.

### Core gene phylogenies and genome features

Genome sequencing and assembly of the test strains yielded genome sizes ranging from 4.64 to 5.02 Mbp with the DNA G + C content ranging from 55.2 to 56.4 mol% (Table S1). The 16S rRNA gene sequences extracted from the genome sequences exhibited 100% identity with those generated by Sanger sequencing.

The phylogenetic analysis based on *bac120* genes placed all the test strains in a single cluster that included all *Enterobacter* species (Fig. 1), confirming their affiliation with the genus *Enterobacter*. Among the test strains, three mutant strains were closely related to the type strains of *Enterobacter asburiae* and *Enterobacter dykesii*, while strain C45 was placed near the type strain of *E. quasihormaechei*. Strain P99 formed a subcluster with the type strains of *Enterobacter hormaechei* subsp. *steigerwalttii*, which has been recently reclassified as "*Enterobacter xiangfangensis*" (7). However, it is important to note that the *E. asburiae* cluster included the type strain of *E. dykesii*. Similarly, the "*E. xiangfangensis*" cluster, which consists of type strains of *E. hormaechei* subsp. *xiangfangensis*, *E. hormaechei* subsp. *oharae*, and *E. hormaechei* subsp. *steigerwaltii* along with strain P99,

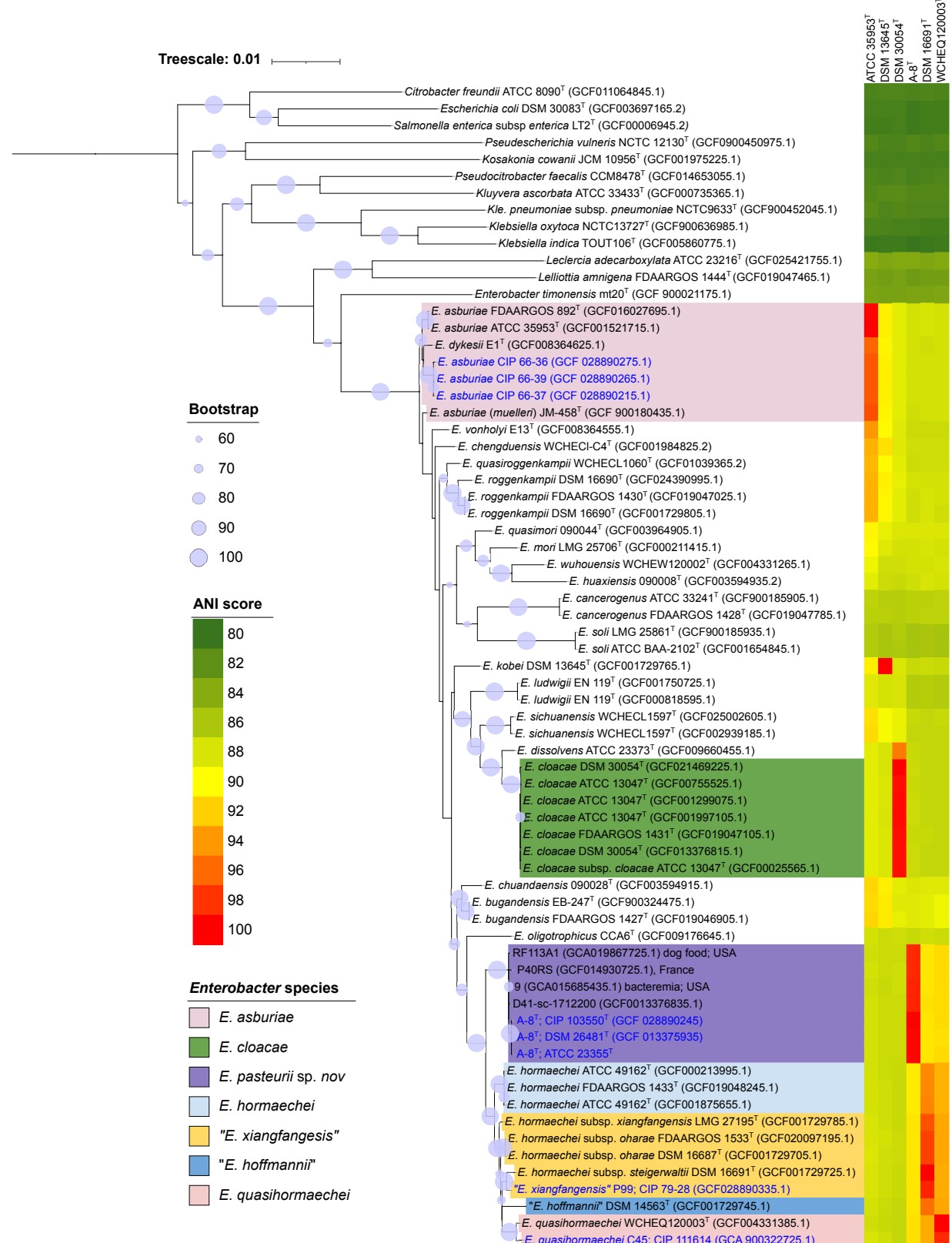

**FIG 1** Phylogenetic tree inferred using the likelihood method in IQ-TREE pipeline using the concatenated alignment of bac120 genes of six *E. cloacae* complex strains and their closely related taxa. Values shown next to the branches are the percentage of replicate trees with associated taxa clustered together in the bootstrap test (1,000 replicates). The color strips indicate the ANI values.

was not monophyletic. These observations indicated the limitations of bac120 genes in discriminating certain *Enterobacter* species. Furthermore, these clusters of bac120 gene phylogeny under question were also in conflict with ANI results, as the strains with ANI value differences below 95% were placed in the same cluster. Interestingly, the strain A-8$^T$ was placed in a monophyletic cluster devoid of any type strain of previously described species, indicating that it represents a new species.

Pan-genome analysis identified 2,551 genes representing the core-genome of the *Enterobacter* species and the test strains. The phylogenetic analysis based on these core genes provided a monophyletic grouping at the species level for all *Enterobacter* species, which was consistent with the ANI results (Fig. 2). Consequently, the results of the core gene phylogeny and ANI values confirmed that the three mutant strains belong to *E. asburiae*. Notably, the type strain of *E. dykesii* was placed outside *E. asburiae* cluster. Similarly, based on the core gene phylogeny and ANI values, strain C45 was identified as a member of the recently described *E. quasihormaechei* (6). Furthermore, the strain P99 was assigned to "*E. xiangfangensis*," as it was placed in the cluster of "*E. xiangfangensis*" including all previously classified subspecies of *E. hormaechei* (34). However, a recent study suggested that *E. hormaechei* subsp. *steigerwaltii*, *E. hormaechei* subsp. *oharae*, and *E. xiangfangensis* subsp. *xiangfangensis* should not be considered subspecies of *E. hormaechei* and are reclassified to "*E. xiangfangensis*" as they are closely related (7). We also proposed that *E. hormaechei* subsp. *steigerwaltii* and *E. hormaechei* subsp. *oharae* should be considered synonyms of "*E. xiangfangensis*" (7). Recently, "*E. xiangfangensis*" has been identified as the predominant clinical species worldwide based on genomic studies and retrospective analysis of the hsp60 gene (35).

In contrast to other strains, the strain A-8$^T$ was placed within a monophyletic cluster in both *bac120*-gene and core gene phylogenies, alongside four strains D41-sc-1712200, P40RS, 9, and RF113A1 (showed more than 98% sequence similarity of *recA* gene of strain A-8$^T$), independent of all described *Enterobacter* species (Fig. 1 and 2). The high ANI and dDDH values, exceeding 99% and 89%, respectively, shared between strain A-8$^T$ and four strains of this cluster, indicate their inclusion into the same species. Moreover, the DNA G + C content exhibited that minimal variation across all the strains of this cluster, with values differing by less than 1% (ranging from 56.1 to 56.7 mol%), provided further confirmation of their classification to the same species (36). In contrast, when compared to all type strains of described *Enterobacter* species, the ANI and dDDH values with the strain A-8$^T$ were found to be below the species threshold, measuring less than 92% and 44%, respectively (37, 38). This clear deviation from the established thresholds of ANI and dDDH confirms that the cluster containing strain A-8$^T$ represents a novel species within the genus *Enterobacter*. Notably, one of the strains (strain D41-sc-1712200) from this cluster has already been identified as one of the 22 unnamed genomospecies of *Enterobacter* (39). Furthermore, it is worth mentioning that the isolates within this cluster have been obtained from human clinical samples and dog food in France and the USA. This information is crucial for understanding the ecology and biogeography of this new species, especially when the source of isolation for strain A-8$^T$ is unknown.

An extensive survey of various culture collections, including CIP, ATCC, DSMZ, and CECT, revealed that strain A-8$^T$ was initially deposited by the Department of Medical Microbiology at Stanford University as *A. cloacae* in ATCC, and subsequently distributed to CIP. At present, the strain A-8$^T$ is available at several culture collections and has been accessed by researchers across the globe. Likewise to CIP, its genome was sequenced independently by the other culture collections including DSMZ and ATCC. The genome of strain A-8$^T$ (DSM 26481$^T$) was also sequenced by using PacBio-based long-read approach to confirm its taxonomic position (13). Interestingly, bac120 gene and core gene phylogenies and ANI results confirmed that the genome sequences of strain A-8$^T$ that originated from different culture collections (ATCC and DSMZ) are 100% identical to the genome sequence we obtained in this study.

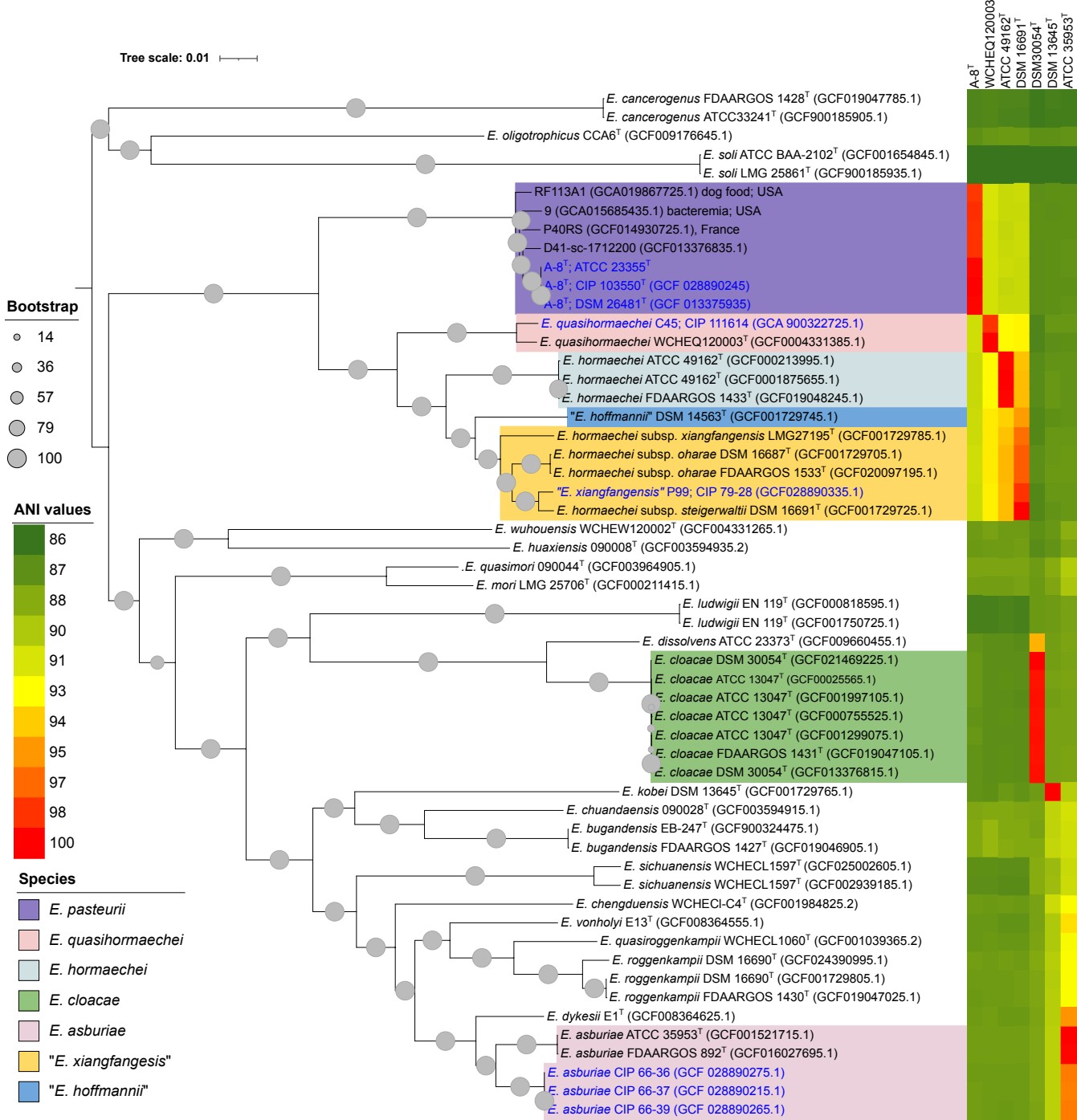

**FIG 2** Phylogenetic tree inferred using the likelihood method in IQ-TREE pipeline using the concatenated sequences of 2,551 core genes identified and extracted from only *Enterobacter* genome data set (consisting six *E. cloacae* complex CIP strains, four strains identical to strain A-8[T], and all type strains of *Enterobacter* species) using Panaroo. Values shown next to the branches are the percentage of replicate trees with associated taxa clustered together in the bootstrap test (1,000 replicates). The color strips indicate the ANI values.

## MALDI-TOF MS-based typing

An analysis of the MALDI-TOF MS spectrum of the three mutant strains in the Bio- typer database resulted in high score values (more than 2.3) with *E. asburiae*. How- ever, the remaining three strains (C45, P99, and A-8[T]) showed scores (more than 2.0)

corresponding to *E. hormaechei*. A score above 2.3 in MALDI- TOF MS biotyper indicates a highly probable species, while the score between 2.0 and 2.29 confirms genus-level identification and probable species identification. The MALDI-TOF biotyper results were consistent with the core gene phylogeny for three mutant strains. However, they did not clearly identify the remaining strains (C45, P99, and A-8[T]) at the species level and only confirm their genus-level identification.

Cluster analysis of the test strains with other members of *Enterobacteriaceae* placed all of them within *Enterobacter* (Fig S2). The three mutant strains are in close proximity to *E. asburiae* CCM 4032, whereas strains C45, P99, and A-8[T] are placed independently and branched from all other strains of *Enterobacter* species at a distance higher than 300. This indicates their high dissimilarity to the spectral profiles present in the biotyper database. A comparison of MALDI-TOF MS spectral profiles of the test strains revealed that strain A-8[T] exhibited 18 unique peaks (Table S2). Furthermore, there were several peaks present in at least one of the test strains that were not detected in strain A-8[T]. These distinctive spectral peaks can be utilized in developing a MALDI-TOF MS biotyper-based diagnosis for this new species represented by strain A-8[T]. Various studies indicate that enrichment of the database with MALDI-TOF MS spectral profiles can substantially improve the identification rate even for *E. cloacae* complex members (30, 40). In addition to facilitating rapid identification, MALDI-TOF MS-based protein profiling can be used for the internal quality control of microbial resource at mBRCs (29).

## Phenotypic characterization of strain A-8[T]

Strain A-8[T] demonstrated robust growth on various general media, including nutrient agar, TSA, and brain heart infusion (BHI) agar. After 24 h of incubation at 37°C on TSA, colonies of strain A-8[T] appeared as circular, translucent, raised form with irregular margins (Fig. S3). Notably, the strain exhibited sediment aggregation when grown in liquid medium without shaking. The strain grew in a range of temperature between 20°C and 41°C, with optimal growth observed at 37°C. In terms of pH, strain A-8[T] displayed growth across a range of 5.5–11, with the best growth observed at pH 7. Microscopically, the cells appeared heteromorphic, with the majority measuring 1.5–3.0 µm in length, although longer filaments ranging from 8 to 20 µm were also observed (Fig. 3; Fig. S4). Similar morphological plasticity has been observed in several other pathogenic bacteria, including members of the *Enterobacteriaceae* family (41). Additionally,

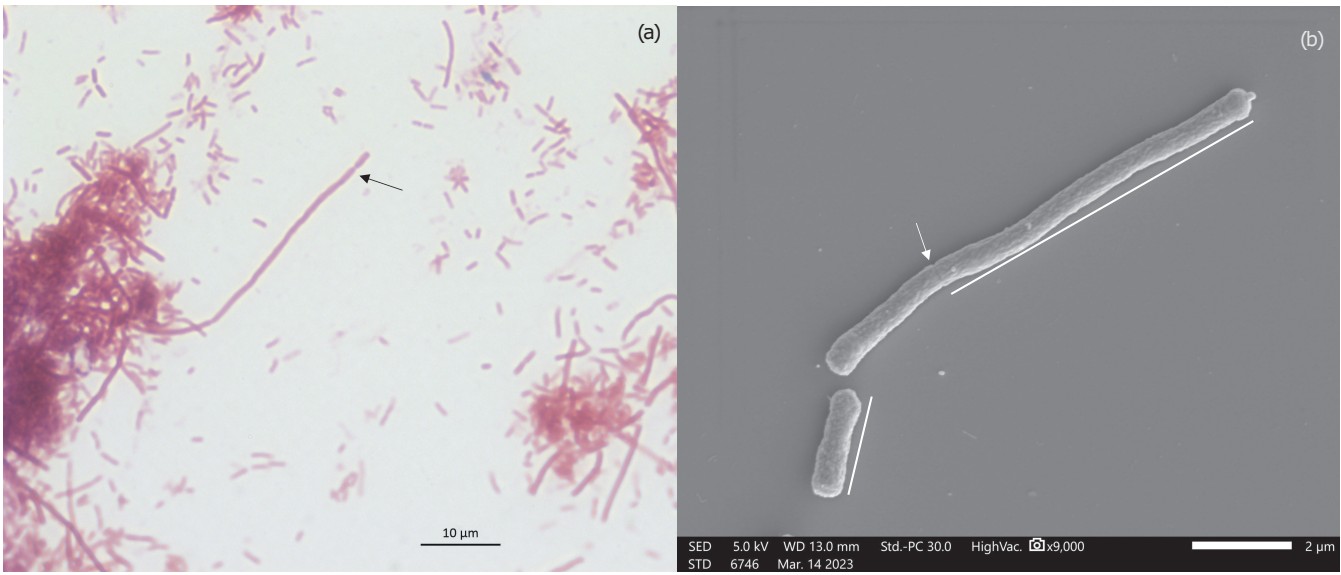

**FIG 3**  (a) Gram stain image of cells of strain A-8[T] under a bright-field microscope at magnification 100×; (b SEM image of cells of strain A-8[T]. Arrows indicating the formation of constriction.

bright-field microscopy and SEM revealed the segmentation of these long filaments. Through negative staining and TEM imaging, fimbriae-like structures were observed throughout the cells, while no flagella were detected (Fig. S4). The cells of the strain were Gram-negative and non-motile (when tested in semi-solid media). This is atypical of *Enterobacter*, as majority of *Enterobacter* species are motile, with the exception of *Enterobacter sichuanensis* and *E. asburiae* (42, 43). Strain A-8[T] exhibited good growth under anaerobic conditions. It tested negative for oxidase but positive for catalase and nitrate reductase. Further details regarding the biochemical profile of strain A-8[T] and closely related taxa are provided in Table 2; Table S3a through c.

## Antimicrobial resistance genes and virulence factors

In Table S4, the search of the CARD database revealed 22 antimicrobial resistance genes with more than 80% sequence identity in the strain A-8[T] and strains D41-sc-1712200, P40RS, 9, and RF113A1 of novel *Enterobacter* species. However, the NCBI and Resfinder databases yielded only four and eight antimicrobial resistance genes, respectively. Among these genes, beta-lactamase ACT (*blaACT*), fosfomycin resistance glutathione transferase (*fosA*), and multidrug efflux RND transporters (*oqxA* and *oqxB*) were consistently found in all strains belonging to the novel *Enterobacter* species. It is worth mentioning that *blaACT* genes were the most dominant beta-lactamase genes found among the members of the *E. cloacae* complex (2). Additionally, homologs of *fosA*, which contribute to intrinsic fosfomycin resistance (51), were present in the majority of genomes of several *Enterobacteriaceae* species including *E. cloacae*.

Notably, strain A-8[T] exhibited resistance to erythromycin, pristinamycin, and fosfomycine in the disc diffusion assay (Table S3d) as per the recommendations of CA-SFM/EUCAST (32), corroborating the presence of corresponding antibiotic resistance genes in its genome.

In addition to antimicrobial resistance, the virulence factor database search identified a total of 33 unique virulence factor genes in the novel *Enterobacter* species, displaying

**TABLE 2** Biochemical characteristics of *E. pasteurii* sp. nov. strain A-8[T] and the type strains of *Enterobacter* species

| Characteristic | 1[a] | 2 | 3 | 4 | 5 | 6 | 7 | 8 | 9 | 10 | 11 | 12 | 13 | 14 | 15 | 16 | 17 | 18 | 19 | 20 | 21 |
|---|---|---|---|---|---|---|---|---|---|---|---|---|---|---|---|---|---|---|---|---|---|
| Motility | − | + | + | + | + | + | + | + | + | + | − | + | + | + | + | + | + | − | + | ND | ND |
| Citrate utilization | + | + | + | + | + | + | + | + | + | + | + | + | + | + | + | + | + | + | + | + | + |
| H₂S production | − | − | − | − | − | − | − | − | − | − | − | − | − | − | − | − | − | − | − | − | − |
| Urea hydrolysis | − | − | − | − | − | − | − | − | − | − | − | − | − | − | − | − | − | − | − | − | − |
| Deaminase | − | − | − | − | − | − | − | − | − | − | − | − | + | − | − | − | − | − | − | − | − |
| Indole production | − | − | − | − | − | − | − | − | − | − | − | − | − | − | − | − | − | − | − | − | − |
| Voges-Proskauer reaction | + | − | + | + | + | + | + | + | + | + | + | − | W | + | + | + | + | − | + | + | − |
| β-Galactosidase | + | + | + | + | + | + | + | + | + | + | + | + | + | + | + | + | + | + | + | − | + |
| d-Glucose | + | + | + | + | + | + | + | + | + | + | + | + | + | + | + | + | + | + | + | + | + |
| d-Mannitol | + | + | + | + | + | + | + | + | + | + | − | + | + | + | + | + | + | + | + | + | + |
| Inositol | + | − | − | − | − | − | + | + | − | − | + | W | − | + | − | + | − | − | + | − | − |
| d-Sorbitol | + | − | + | + | − | + | + | + | + | − | + | + | + | + | + | + | − | + | + | + | + |
| l-Rhamnose | + | + | + | + | + | + | + | + | + | + | − | + | + | + | + | + | − | + | + | + | + |
| Sucrose | + | + | + | + | + | + | + | + | + | + | + | + | + | + | + | + | − | + | + | − | − |
| Melibiose | + | − | + | + | + | + | + | + | + | − | + | + | + | + | + | + | − | − | + | + | − |
| Amygdalin | − | + | + | + | + | + | + | + | + | + | + | + | + | + | + | + | + | + | + | + | + |
| Arabinose | + | + | + | + | + | + | + | + | + | + | + | + | + | + | + | + | + | + | + | + | + |
| l-Fucose | − | + | − | − | − | − | − | − | − | − | − | − | ND | V | − | V | + | − | + | ND | + |
| d-Arabitol | − | − | − | − | − | − | + | − | − | − | − | − | − | + | − | − | − | − | − | ND | − |
| Dulcitol | − | + | − | W | − | − | − | + | − | − | − | − | ND | − | + | − | − | − | − | ND | + |
| d-Turanose | − | + | − | − | ND | ND | − | + | − | − | − | + | ND | + | − | − | − | W | W | ND | − |

[a]Species: 1, *E. pasteurii* sp. nov.; 2, *E. hormaechei*; 3, "*E. xiangfangensis*"; 4, *E. cloacae*; 5, *E. quasihormaechei*; 6, *E. wuhouensis*; 7, *E. quasiroggenkampii*; 8, *E. quasimori*; 9, *E. huaxiensis*; 10, *E. chuandaensis*; 11, *E. sichuanensis*; 12, *E. chengduensis*; 13, *Enterobacter soli*; 14, *Enterobacter mori*; 15, *Enterobacter bugandensis*; 16, *Enterobacter ludwigii*; 17, *Enterobacter cancerogenus*; 18, *E. asburiae*; 19, *Enterobacter kobei*; 20, *Enterobacter timonensis*; and 21, *Enterobacter oligotrophica*. Data for *Enterobacter* species, except the newly described one, are from references (6, 7, 43–50). ND, not determined; V, varied; W, weakly positive.

more than 75% sequence identity (Table S5). Many of these virulence factor genes are implicated in iron uptake or transport, flagella or pili biosynthesis, and chemotaxis, which might have potential involvement in pathogenicity mechanisms (1). Furthermore, the presence of curli-encoding genes like *csgB*, *csgD*, *csgE*, *csgF*, and *csgG* in the genomes of strains exhibited their potential to adhere and invade the host cell.

## Comparative genome analysis

A total of 748 genes were identified to be present uniquely in the genomes of strains representing the novel *Enterobacter* species when compared with type strains of all *Enterobacter* species (Table S6). Of these, 57 genes were found in the genome of all strains of the proposed novel *Enterobacter* species. A large majority of these genes represent flagellum biosynthesis and function genes. This is contrary to TEM results, where we did not find any flagellum for the strain A-8$^T$ (Fig S4). Additionally, several hypothetical protein genes were also recorded in the genome of these strains. Notably, various pili/fimbriae-related genes that are implicated in adhesion and biofilm formation were uniquely present in the genome of strains representing the novel *Enterobacter* species. This is in confirmation to the presence of several fimbriae-like structures at the surface of strain A-8$^T$ cells detected by electron microscopy (Fig. S4).

The anvi'o pangenomics workflow of *Enterobacter* pan-genome resulted in 14,270 gene clusters, of which 105 genes represent the unique core and 648 gene clusters represent singletons of the newly described species (Fig. 4; Table S7 and S8). The gene cluster specific to the new species includes transcriptional regulator, N-acetylglutamate synthase, pili, flagella, autotransporter, NADPH-quinone reductase, amino acid permease, type III secretory pathway, glycosyltransferase, cell division protein, and several other undefined gene clusters (Table S7).

Notably, the majority of these gene clusters are related to cell motility, intracellular trafficking, secretion, and vesicular transport, possibly enabling the members of this species to colonize the host. Furthermore, the clinical origin of certain strains of this species indicates the relevance of a wide repertoire of virulence factors in providing important adaptive mechanisms.

## Conclusions

Based on detailed genome sequence analyses, we successfully achieved species-level assignments for *Enterobacter* strains. Our analyses also aligned with recently proposed reclassifications within the genus *Enterobacter*. The application of genome sequence data, alongside MALDI-TOF MS biotyper-based protein profiling offers a comprehensive strategy for accurate bacterial strain identification, further contributing to the effective management of mBRCs. By employing a polyphasic approach including genotypic and phenotypic analyses, we present compelling evidence that strain A-8$^T$ represents a new species within the genus *Enterobacter*, for which we propose the name *Enterobacter pasteurii* sp. nov.

## Description of *E. pasteurii* sp. nov

*E. pasteurii* (pas.teu'ri.i. N.L. gen. masc. n. *pasteurii*, in honor of Louis Pasteur, a French microbiologist, who made seminal contributions to the field of infectious diseases).

Cells are facultative aerobic, Gram stain-negative, heteromorphic rods, with majority of cell 1.5–3.0 µm in length, a few longer filaments ranging from 8 to 20 µm, and a width of 0.3–0.7 µm. The growth temperature range is 20°C–41°C (optimum growth at 37°C), and pH range is 5.5–11 (optimum growth at pH 7). Negative is for oxidase, H$_2$S production, urease, and indole production reaction. Positive is for catalase, nitrate reductase, citrate utilization, fermentation of glucose and lactose, malonate utilization, and Voges-Proskauer reaction. D-Glucose, D-fructose, D-galactose, D-trehalose, D-mannose, D-melibiose, sucrose, D-raffinose, maltotriose, maltose, lactose, lactulose, 1-O-methyl-beta-galactopyranoside, D-cellobiose, gentiobiose, 1-O-methyl-beta-D-glucopyranoside, esculin, D-ribose, L-arabinose, D-xylose,

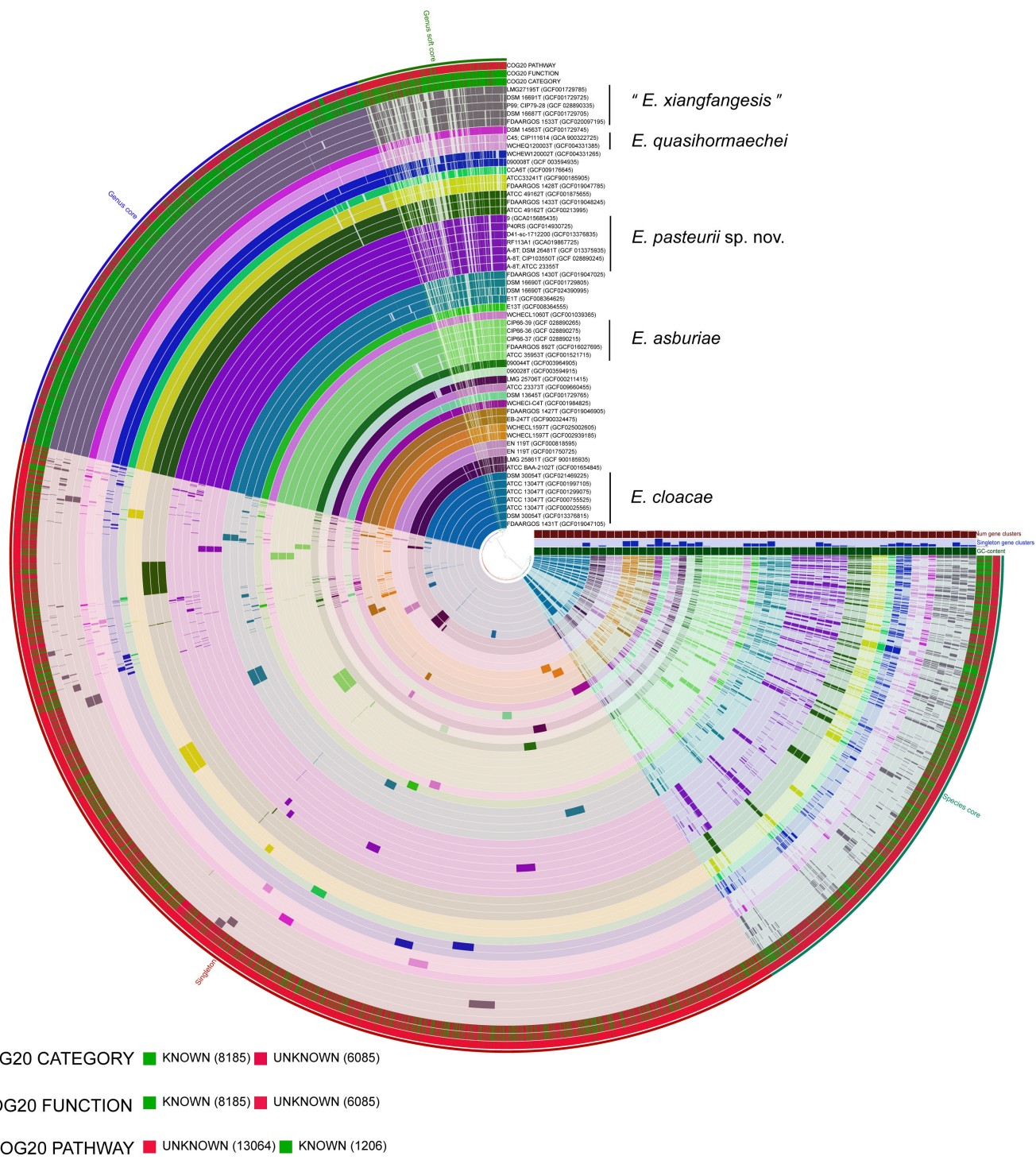

**FIG 4** Pangenome analysis depicting the shared and unique gene clusters (nos. 14,270) among the members of the genus *Enterobacter*. The genomes are organized in radial layers such as genus core, genus soft core, species core, and unique singleton gene clusters (Euclidean distance; Ward linkage) which are defined by the gene tree in the center. *E. pasteurii* sp. nov. is represented by genomes including strain A-8[T] and four strains identical to A-8[T], while *E. asburiae*, *E. quasihormaechei*, and "*E. xiangfangesis*" include genomes of strains used in this study along with the type strains, and the remaining species are represented by genomes of their type strains.

L-rhamnose, glycerol, myo-inositol, D-mannitol, D-sorbitol, D-saccharate, mucate, D-malate, L-malate, *cis*-aconitate, *trans*-aconitate, citrate, D-glucuronate, D-galacturonate, 2-ketogluconate, N-acetyl-D-glucosamine, D-gluconate, phenylacetate, DL-lactate,

L-histidine, succinate, fumarate, D-glucosamine, L-aspartate, L-glutamate, D-alanine, L-alanine, L-serine, and malonate were utilized on Biotype-100 strips. Negative is for the utilization of L-sorbose, 1-O-methyl-alpha-galactopyranoside, palatinose, L-fucose, D-melezitose, D-arabitol, L-arabitol, xylitol, dulcitol, D-tagatose, maltitol, D-turanose, adonitol, HQ-B-glucuronide, D-lyxose, i-erythritol, 1-O-methyl-alpha-D-glucopyrano-side, 3-O-methyl-D-glucopyranose, L-tartrate, D-tartrate, meso-tartrate, tricarballylate, 5-ketogluconate, tryptophan, protocatechuate, 4-hydroxybenzoate, quinate, gentisate, 3-hydroxybenzoate, benzoate, 3-phenylpropionate, m-coumarate, trigonelline, betaine, putrescine, 4-aminobutyrate, histamine, caprate, caprylate, glutarate, DL-glycerate, 5-aminovalerate, ethanolamine, tryptamine, itaconate, 3-hydroxybutyrate, L-proline, propionate, L-tyrosine, and 2-ketoglutarate on Biotype-100 strips. Positive is for alkaline phosphatase, esterase (C4), lipase esterase (C8), leucine arylamidase, acid phosphatase, naphtol-AS-BI-phosphohydrolase, alpha-galactosidase, beta-galactosidase, alpha-gluco-sidase, beta-glucosidase, and N-acetyl-beta-glucosaminidase on APIzym strips.

The source of isolation for the type strain A-8[T] (CIP 103550[T]; ATCC 23355[T]; CCUG 33777[T]; CECT 5075[T]; WDCM 00082[T]; DSM 26481[T]; and NCTC 13380[T]) is unknown. However, the strain A-8[T] has been widely used as a standard strain for quality control and to study microbe-microbe interactions. The genome sequences of the strain A-8[T] (CIP 103550[T]) are deposited in NCBI GenBank with the number JARCHI000000000 (GCF_028890245.1). Additionally, strains RF113A1 (GCA_019867725.1), 9 (GCA_015685435.1), D41-sc-1712200 (GCF_013376835.1), and P40R (GCF_014930725.1) are also members of this species and have been isolated from different sources, such as human clinical samples and dog food.

## ACKNOWLEDGMENTS

We thank the Mutualized Platform for Microbiology (P2M) of Institut Pasteur for next-generation sequencing. The computational and storage services offered by the High Performance Computing Core Facility at Institut Pasteur greatly facilitated this work. We extend our thanks to Dr. Federica Palma and Mr. Martin Boutroux for their insightful discussions on genomic analysis and Dr. Adriana Chiarelli for discussions on antimicrobial resistance. Additionally, we acknowledge Dr. Mariana Ferrari for her meticulous review of this manuscript.

We thank the project HoloZcan (grant agreement No. 101021723) of the European Union's Horizon 2020 Research and Innovation program. We are grateful for support for equipment from the French Government Programme Investissements d'Avenir France BioImaging (FBI, N° ANR-10-INSB-04–01) and the French government (Agence Natio-nale de la Recherche) Investissement d'Avenir programme, Laboratoire d'Excellence "Integrative Biology of Emerging Infectious Diseases" (ANR-10-LABX-62-IBEID).

P.R. conceived the study, designed and performed the experiments, analyzed the results, and wrote the manuscript. E.M. performed the preliminary data analysis. C.S. performed the electron microscopy. G.T. performed the anaerobic growth of the bacterium. D.C. conceptualized the study and checked the manuscript.

## AUTHOR AFFILIATIONS

[1]Collection of Institut Pasteur (CIP), Institut Pasteur, Université Paris Cité, Paris, France
[2]Ultrastructural Bioimaging Unit, Institut Pasteur, Université Paris Cité, Paris, France

## AUTHOR ORCIDs

Praveen Rahi  http://orcid.org/0000-0002-3154-9616
Cyril Scandola  http://orcid.org/0000-0002-5305-9095

## FUNDING

| Funder | Grant(s) | Author(s) |
|---|---|---|
| EC \| Horizon Europe \| 創新的歐洲 \| HORIZON EUROPE European Innovation Council (EIC) | 101021723 | Praveen Rahi |
| | | Dominique Clermont |
| Agence Nationale de la Recherche (ANR) | ANR-10-INSB-04-01 | Cyril Scandola |

## AUTHOR CONTRIBUTIONS

Praveen Rahi, Conceptualization, Data curation, Formal analysis, Investigation, Methodology, Project administration, Resources, Supervision, Validation, Visualization, Writing – original draft, Writing – review and editing | Estelle Mühle, Formal analysis, Methodology, Validation | Cyril Scandola, Formal analysis, Methodology, Writing – review and editing | Gerald Touak, Formal analysis, Methodology | Dominique Clermont, Conceptualization, Project administration, Supervision, Writing – review and editing

## DATA AVAILABILITY

Genome sequence data have been deposited in NCBI under the following accession numbers: A-8$^T$ GCF_028890245.1 (JARCHI000000000); P99 GCF_028890335.1 (JARCHJ000000000); C45 GCA_900322725.1 (LT991957.1); CIP 66.36 GCF_028890275.1 (JARCHK000000000); CIP 66.37 GCF_028890215.1 (JARCHL000000000); and CIP 66.39 GCF_028890265.1 (JARCHM000000000).

## ETHICAL APPROVAL

The bacterial strains reported in the study were obtained from the Collection of Institut Pasteur and do not involve any work on or with animals.

## ADDITIONAL FILES

The following material is available online.

### Supplemental Material

**Supplemental figures (Spectrum03150-23-s0001.pdf).** Fig. S1 to S4.
**Supplemental tables (Spectrum03150-23-s0002.xlsx).** Tables S1 to S8.

### Open Peer Review

**PEER REVIEW HISTORY (review-history.pdf).** An accounting of the reviewer comments and feedback.

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
