## [Reviewer comments · Microbiology Spectrum]

Microbiology Spectrum

Genome Sequence-Based Identification of *Enterobacter* Strains and Description of *Enterobacter pasteurii* sp. nov.

Praveen Rahi, Estelle Mühle, Cyril Scandola, Gerald Touak, and Dominique Clermont

Corresponding Author(s): Praveen Rahi, Institut Pasteur

Review Timeline:

Submission Date:	August 22, 2023
Editorial Decision:	October 21, 2023
Revision Received:	October 31, 2023
Accepted:	November 16, 2023

Editor: Shannon Manning

Reviewer(s): The reviewers have opted to remain anonymous.

Transaction Report:

DOI: <https://doi.org/10.1128/spectrum.03150-23>

October 21, 2023

Dr. Praveen Rahi
Institut Pasteur
Collection of Institut Pasteur
Institut Pasteur, Université Paris Cité
Paris 75015
France

Re: Spectrum03150-23 (Genome Sequence-Based Identification of *Enterobacter* Strains and Description of *Enterobacter pasteurii* sp. nov.)

Dear Dr. Praveen Rahi:

Thank you for submitting your manuscript to Microbiology Spectrum. We have received reports from 2 reviewers and would like to invite you to revise your work for further consideration. While we agree that the work is important and the manuscript is well written, the Reviewer's have several questions that need to be addressed. Reviewer 1 has requested additional genomic information, while Reviewer 2 indicated that the novel taxa should be officially approved and verified.

Link Not Available

Sincerely,

Shannon Manning

Journals Department
Reviewer comments:

Reviewer #1 (Comments for the Author):

This paper is a very well-structured investigation which leads to the description of a novel species of Enterobacteria, i.e. *E. pasteurii*. The methods used are all highly appropriate and the study has been very well executed and is of high quality. It would be good if the authors could however provide a few more data and do some minor changes:

1) Please provide a mol% GC content of the genome of the novel type strain in the species description. 2) Please perform a

dDDH analysis of the type strain with the most related strains and include the results in the text.

Minor changes:

line 59: ...oldest culture collections and...(add s to collection/plural) include "and"

Insert spaces between heading/section i.e. lines 122/123, 164/165, 213/214, 274/275367/368, 376/377

line 144: Is there a specific software version of panaroo used?

Line 187: is it Tryptone Soya broth or Tryptone Soy broth? With "a" after the "y" or not? Please check

line 203: Sanger method with a length (delete s from method, include "a")

line 206: there is growing evidence (replace are with is)

line 239...was placed outside of the *E. asburia* cluster OR: apart from the *E. asburia* cluster

Line 285: Enterobacteriaceae: iaceae not eacea, please correct

line 329 onwards: please write all gene names with small letters and italics: blaACT, fosA,

Write all antibiotic names with small beginning letters: fosfomycin not Fosfomycin, erythromycin not Erythromycin etc...

write all gene names in line 341 in italics

line 383 and onwards, write all enzyme and sugar names with small beginning letters...urease not Urease, D-mannose nor D-Mannose etc..

Finally: the sentence starting in line 406 which intends to mention the source of the type strain is incomplete...the source is not mentioned!

Reviewer #2 (Comments for the Author):

In the manuscript "Genome sequence-based identification of *Enterobacter* strains and description of *Enterobacter pasteurii* sp. nov." (Spectrum03150-23), Rahi et al. sought to apply taxonomic clarity to six *Enterobacter cloacae* complex archival isolates from Institut Pasteur which were previously curated as *Aerobacter aerogenes* (n = 3), beta-lactamase-producing *Enterobacter cloacae* (n = 2), and carbapenem-resistant *Enterobacter* spp. The group utilized multiple genotypic and phenotypic approaches. Notably, 16S rRNA gene-based phylogeny failed to group all six isolates in the same monophyletic cluster; secondary phylogenetic approaches either achieved monophyletic clustering with discrepancies in average nucleotide identity data (bac120-based) or achieved species resolution with and correlative average nucleotide identity data (*Enterobacter*-specific core gene phylogeny). As a result of these studies, the group reported one isolate forming a distinctive cluster within all phylogenies and hereby propose taxonomic assignment as *Enterobacter pasteurii* sp. nov.

The manuscript was well written and easy to follow. Specific comments are itemized below:

- 1) The authors are advised that formal acceptance of novel taxa must take place by way of the International Journal of Systematic and Evolutionary Microbiology, either by primary publication or incorporation by that Journal into Validation Lists;
- 2) Throughout the manuscript, authors are encouraged to verify current official taxonomic status of "*Enterobacter xiangfangensis*" (*Enterobacter hormaechei* subsp. *xiangfangensis*) through the LPSN web resource;
- 3) A re-write of the sentence in line 62 may be indicated;
- 4) Line 68, a reference for the ESKAPE group statement may be indicated;
- 5) Line 72, at time of this manuscript review, 23 current valid and correct species designations within genus *Enterobacter*;
- 6) Line 198 and 333, please provide guidelines utilized for inoculation and interpretation of *Enterobacter* spp. antimicrobial susceptibility testing;
- 7) Line 223, authors are encouraged to verify current official taxonomic status of "*Enterobacter xiangfangensis*" (*Enterobacter hormaechei* subsp. *steigerwaltii* is a current valid and correct designation) through the LPSN web resource;
- 8) Lines 406 and 407, incomplete sentence;
- 9) Can additional clinical relevance data be ascribed to the isolates that are documented in Table 1?
- 10) Are decarboxylase/dihydrolase data available for the isolates in question (Table 2)?

Staff Comments:

Preparing Revision Guidelines

- Point-by-point responses to the issues raised by the reviewers in a file named "Response to Reviewers," NOT IN YOUR COVER LETTER.
- Upload a compare copy of the manuscript (without figures) as a "Marked-Up Manuscript" file.

- Each figure must be uploaded as a separate file, and any multipanel figures must be assembled into one file.
- Manuscript: A .DOC version of the revised manuscript
- Figures: Editable, high-resolution, individual figure files are required at revision, TIFF or EPS files are preferred

Please return the manuscript within 60 days; if you cannot complete the modification within this time period, please contact me. If you do not wish to modify the manuscript and prefer to submit it to another journal, please notify me of your decision immediately so that the manuscript may be formally withdrawn from consideration by Microbiology Spectrum.

Response to reviewers' comments:

Reviewer #1 comments	Authors response
This paper is a very well-structured investigation which leads to the description of a novel species of Enterobacteria, i.e. E. pasteurii. The methods used are all highly appropriate and the study has been very well executed and is of high quality. It would be good if the authors could however provide a few more data and do some minor changes: 1) Please provide a mol% GC content of the genome of the novel type strain in the species description. 2) Please perform a dDDH analysis of the type strain with the most related strains and include the results in the text.	Thank you for your constructive comments and suggestions. We have included the information for DNA G+C content and dDDH analysis in the text.
Minor changes: line 59: ...oldest culture collections and...(add s to collection/plural) include "and"	Correction done.
Insert spaces between heading/section i.e. lines 122/123, 164/165, 213/214, 274/275, 367/368, 376/377	Correction done.
line 144: Is there a specific software version of panaroo used?	Panaroo version 1.3.0
Line 187: is it Tryptone Soya broth or Tryptone Soy broth? With "a" after the "y" or not?	It is Tryptone Soy broth we corrected it.
Please check line 203: Sanger method with a length (delete s from method, include "a") line 206: there is growing evidence (replace are with is) line 239...was placed outside of the E. asburia cluster OR: apart from the E. asburia cluster Line 285: Enterobacteriaceae: iaceae not eacea, please correct line 329 onwards: please write all gene names with small letters and italics: blaACT, fosA, Write all antibiotic names with small beginning letters: fosfomicin not Fosfomicin, erythromycin not Erythromycin etc... write all gene names in line 341 in italics	Thank you for this, all correction done as suggested.

line 383 and onwards, write all enzyme and sugar names with small beginning letters...urease not Urease, D-mannose nor D-Mannose etc..	
Finally: the sentence starting in line 406 which intends to mention the soource of the type strain is incomplete...the source is not mentioned!	Thank you for pointing this, we mentioned that the source of type strain is unknown. However, the strain A-8T has been widely used as a standard strain for quality control and to study microbe-microbe interactions.
Reviewer #2 comments	Authors response
In the manuscript "Genome sequence-based identification of Enterobacter strains and description of Enterobacter pasteurii sp. nov." (Spectrum03150-23), Rahi et al. sought to apply taxonomic clarity to six Enterobacter cloacae complex archival isolates from Institut Pasteur which were previously curated as Aerobacter aerogenes (n = 3), beta-lactamase-producing Enterobacter cloacae (n = 2), and carbapenem-resistant Enterobacter spp. The group utilized multiple genotypic and phenotypic approaches. Notably, 16S rRNA gene-based phylogeny failed to group all six isolates in the same monophyletic cluster; secondary phylogenetic approaches either achieved monophyletic clustering with discrepancies in average nucleotide identity data (bac120-based) or achieved species resolution with and correlative average nucleotide identity data (Enterobacter-specific core gene phylogeny). As a result of these studies, the group reported one isolate forming a distinctive cluster within all phylogenies and hereby propose taxonomic assignment as Enterobacter pasteurii sp. nov. The manuscript was well written and easy to follow.	Thank you for your constructive comments and suggestions.
Specific comments are itemized below: 1) The authors are advised that formal acceptance of novel taxa must take place by way of the International Journal of Systematic and Evolutionary Microbiology, either by primary publication or	Thank you for your suggestion, and yes, we are aware of the validation process. Once the manuscript is accepted and available online, we will make a request with the list editors of IJSEM for the validation of Enterobacter pasteurii.

incorporation by that Journal into Validation Lists	
2) Throughout the manuscript, authors are encouraged to verify current official taxonomic status of "Enterobacter xiangfangensis" (Enterobacter hormaechei subsp. xiangfangensis) through the LPSN web resource	We checked with LPSN and found that " Enterobacter xiangfangensis " is a basonym of Enterobacter hormaechei subsp. xiangfangensis . Although several recent studies on Enterobacter cloacae complex concluded that three sub-species of Enterobacter hormaechei are " Enterobacter xiangfangensis " and our analyses also confirm this. We checked the complete text and all tables and figures for correct expression of " Enterobacter xiangfangensis ", which await the validation of reclassification.
3) A re-write of the sentence in line 62 may be indicated	Thank you pointing this, we re-write this sentence. Many of the strains within the CIP collection were deposited prior to the advent of molecular methods. Consequently, there is a need for a contemporary taxonomic validation process employing state-of-the-art methodologies for such strains.
Line 68, a reference for the ESKAPE group statement may be indicated	Reference for the ESKAPE group statement was added. World Health Organization. 2018. Global antimicrobial resistance surveillance system (GLASS) report: early implementation 2017-2018. World Health Organization, Geneva. https://iris.who.int/handle/10665/279656 . Retrieved 24 October 2023.
5) Line 72, at time of this manuscript review, 23 current valid and correct species designations within genus Enterobacter	Corrected in the text, after confirming with LPSN, Enterobacter nematophilus Machado et al. 2023 is a new species of Enterobacter.
Line 198 and 333, please provide guidelines utilized for inoculation and interpretation of Enterobacter spp. antimicrobial susceptibility testing	Following the recommendations of CA-SFM/EUCAST. Reference provided.
7) Line 223, authors are encouraged to verify current official taxonomic status of "Enterobacter xiangfangensis" (Enterobacter hormaechei subsp. steigerwaltii is a current valid and correct designation) through the LPSN web resource	We agree that reclassification of the three Enterobacter hormaechei subspecies was not yet validated, therefore, throughout the manuscript we are using " Enterobacter xiangfangensis ".
8) Lines 406 and 407, incomplete sentence	Thank you for pointing this, we have

	completed the sentence.
9) Can additional clinical relevance data be ascribed to the isolates that are documented in Table 1?	We have already included the history of strains and also provided additional information by citing proper references.
10) Are decarboxylase/dihydrolase data available for the isolates in question (Table 2)?	We have included all relevant phenotypic feature data in the supplementary table. We noticed that none of the phenotypic test has decarboxylase/ dihydrolase data.

Re: Spectrum03150-23R1 (Genome Sequence-Based Identification of *Enterobacter* Strains and Description of *Enterobacter pasteurii* sp. nov.)

Dear Dr. Praveen Rahi:

I am pleased to inform you that your manuscript has been accepted for publication in Microbiology Spectrum. I will now forward it to the ASM production staff for publication. Your paper will first be checked to make sure all elements meet the technical requirements. ASM staff will contact you if anything needs to be revised before copyediting and production can begin. Otherwise, you will be notified when your proofs are ready to be viewed.

Thank you for the opportunity to review your work.

Sincerely,
Shannon Manning
Editor
Microbiology Spectrum

Reviewer #1 (Comments for the Author):

All my previous comments were addressed satisfactory.

Reviewer #2 (Comments for the Author):

The authors have sufficiently addressed previous Reviewers' concerns.